# The HER2MtGx Metagene Score as a Reliable Tool to Select HER2 Breast Cancer Patients for Neoadjuvant Targeted Therapy

**DOI:** 10.3390/ijms262411809

**Published:** 2025-12-06

**Authors:** Daniel Guimarães Tiezzi, Isabela Panzeri Carlotti Buzatto, Willian Abraham da Silveira, Anna Clara Monti, Fabiana de Oliveira Buono, Juliana Meola, Omero Benedicto Poli-Neto, Stefano Maria Pagnotta

**Affiliations:** 1Laboratory for Translational Data Science, Breast Disease Division, Department of Gynecology and Obstetrics, Ribeirão Preto Medical School, University of São Paulo, Ribeirão Preto 14048-900, Brazil; ipcarlotti@hcrp.usp.br (I.P.C.B.); buono@alumni.usp.br (F.d.O.B.); jumeola@usp.br (J.M.); polineto@usp.br (O.B.P.-N.); 2School of Science, Engineering and Environment, University of Salford, Salford M5 4WT, UK; w.a.dasilveira@salford.ac.uk; 3Dipartimento di Diritto, Economia, Management e Metodi Quantitativi, Università degli Studi del Sannio, 82100 Benevento, Italy; acmonti@unisannio.it; 4Dipartimento di Scienze e Tecnologie, Università degli Studi del Sannio, 82100 Benevento, Italy; pagnotta@unisannio.it

**Keywords:** breast cancer, molecular profile, predictive factors, targeted therapy

## Abstract

The cHER2+ breast cancer subtype is characterized by the overexpression of the HER2 oncoprotein based on immunohistochemistry (IHC)/or by *ERBB2* gene amplification using in situ hybridization (ISH) techniques. Targeted therapies are significantly changing cancer treatment outcomes. However, not all patients benefit from it due to misclassification or intrinsic mechanisms of resistance. Identifying predictive factors of response to therapy is thus crucial for optimizing treatment protocol. In addition, with the development of effective antibody–drug conjugates for targeting HER2-low subtypes, enhancing the HER2 molecular classification is crucial. In this study, a comprehensive analysis of publicly available datasets (TCGA, METABRIC, I-SPY, NOAH and CHER-LOB trials) has been considered. We present a metagene expression score (HER2MtGx 31-gene assay) based on the most informative genes associated with each molecular profile. HER2MtGx scores represent three linear subspaces associated with the HER2, Luminal and Basal-like profiles (STAT). In the METABRIC cohort, the scores are useful to discriminate against the HER2-enriched phenotype and this classification is significantly associated with long-term survival in cHER2+ patients (HR = 1.76; 95%CI = 1.09–2.86). In terms of response to neoadjuvant chemo/target therapy including I-SPY, NOAH, and CHER-LOB trials, the metagene scores are associated with the pathological response to therapy (OR = 2.26; 95%CI = 1.74–2.98). The HER2MtGx assay is a reliable tool for selecting patients for HER2-targeted therapy.

## 1. Introduction

Breast cancer is a heterogeneous disease, and there are at least five molecular subtypes associated with distinct clinical and biological behaviors: Luminal A, Luminal B, HER2-enriched (HER2+), normal-like, and Basal-like subtypes. This classification is based on an unsupervised clustering analysis from transcriptomic data [1]. In clinical practice, the HER2+ subtype (cHER2+) is characterized by the overexpression of the HER2 oncoprotein based on immunohistochemistry (IHC) analysis or by the amplification of the ERBB2 gene using in situ hybridization techniques [2]. There is a substantial discordance in predicting the molecular subtype by IHC in cHER2+ tumors [3].

cHER2+ is described as an aggressive breast cancer subtype, and all patients are candidates to receive targeted therapy. Adding adjuvant trastuzumab to treat early-stage cHER2+ patients adds a benefit of 9% in 5-year disease-free survival [4]. In the Brazilian cohort, including early and locally advanced stages, we estimate a benefit of 14% in overall survival in a real-world study [5]. So, not all patients selected to receive HER2-targeted therapy benefit from it. In the precision medicine era and with the development of effective antibody–drug conjugates for targeting HER2-low subtypes [6], good characterization of the molecular alterations underlying the HER2 phenotype is essential for targeted therapy optimization.

The lack of benefit from delivering targeted therapy may be due to bias in the patient selection (i.e., misinterpretation of the IHC) or an intrinsic mechanism of resistance to therapy (i.e., intratumor heterogeneity). Currently, several HER2 target drugs are available in the neoadjuvant and adjuvant setting, each associated with different costs, toxic effects, and rates of pathologic complete response (pCR). The pCR rate, for instance, ranges from 6 to 80%, depending on the therapeutic regimen and ER/PR status [7]. Thus, identifying predictive factors of response to therapy is crucial to determining the best cost-effective treatment protocol. Integrating the molecular and genomic data to better understand the cHER2+ breast cancer heterogeneity may give new insights to optimize cHER2+ breast cancer treatment.

## 2. Results

### 2.1. TCGA HER2 Breast Cancer Reclassification

There are 90 3+ samples and 199 2+ samples based on the HER2 IHC/FISH, and a total of 113 samples were considered positive after reclassification. Table 1 summarizes the distribution of pathological and molecular features after reclassification. The molecular profile makes clear that the cHER2+ samples are highly heterogeneous, and most cHER2+ samples are not classified as HER2-enriched based on the PAM50 classification and are not classified as iC5, a genomic subtype characterized by the amplification in the 17q12-q21 region. These observations reinforce the need to optimize the classification of cHER2+ breast carcinomas.

### 2.2. Molecular Profile of Potentially HER2 Positive Breast Cancer

The 287 potential HER2-positive samples were subjected to unsupervised hierarchical clustering to explore the possible segregation of samples in homogeneous groups according to gene expression levels. AWST [8] protocol identified five clusters (Figure 1). Cluster 5 is highly associated with HER2 expression by IHC and the HER2 molecular profile. A total of 47% of the cHER2+ samples did not match cluster 5. Cluster 3 is the Basal-like samples (triple negative profile at IHC), and clusters 1, 2 and 4 represent three distinct molecular subtypes of the ER-positive Luminal samples.

The copy number analysis of all genes located at the chr17q21-q21 region (Figure 2) demonstrates that the molecular profile classification accurately captures the genomic amplification associated with the HER2-enriched subtype (cluster 5). There is a consistent core amplification associated with cluster 5 encompassing nine genes besides the *ERBB2* (*CDK12*, *NEUROD2*, *PPP1R1B*, *STARD3*, *TCAP*, *PNMT*, *PGAP3*, *MIR4728*, *MIEN1*, *GRB7*, *IKZF3*, *ZPBP2* and *GSDMB*). The average of this core amplicon copy number (log2 ratio) is significantly higher in cluster 5 (F = 34.38, *p* < 0.0001; post hoc Dunn’s test in Appendix A).

### 2.3. Selecting the Most Representative Genes

A total of 347 genes were identified to predict the HER2, Basal-like, and Luminal signatures. After filtering by the highest signal in each signature, a total of 31 genes were selected to build the metagene scores: HER2 genes—*ERBB2*, *PGAP3*, *GRB7*, *STARD3*, *CREB3L1*, *TMEM86A*, *CBX2*, *SOX11*; Basal genes—*FOXA1*, *ROPN1*, *SCRG1*, *DSG1*, *PRSS33*, *GDF5*, *CCKBR*; Luminal genes—*SLC39A6*, *TFF1*, *AGR3*, *GFRA1*, *ESR1*, *SCUBE2*, *RERG*, *THSD4*, *KIF12*, *CCDC74A*, *AFF3*, *CLSTN2*, *BCL2*, *MAPT*, *KCNK15, WNK4*.

The metagene score for each module (HER2, Basal, and Luminal) is calculated by the weighted mean expression using the ANOVA coefficients. The coefficients are kept confidential for patent purposes. The HER2MtGx assay shows that the metagene scores provide the molecular weights associated with each modules in all TCGA samples (Figure 3A–C).

### 2.4. HER2 Metagene Score Impact on Breast Cancer-Specific Survival

The HER2MtGx score shows a reliable generalization from the RNAseq data to a microarray data. There is a significant association between the metagene scores and the intClust subtypes (*p* < 0.0001) in the METABRIC dataset, and it is able to capture the iC5 subtype (Figure 4A).

To evaluate the impact on survival, 230 patients with non-metastatic cHER2+ patients were stratified into HER2-low and HER2-high groups based on the median HER2 metagene score. Tumor stage (stage 2: HR = 2.2 [1.2–3.8], *p* = 0.006; stage 3: HR = 4.4 [2.1–8.3], *p* < 0.0001), ER status (negative: HR = 1.3 [0.96–2], *p* = 0.08), and HER2 metagene score (positive: HR = 1.4 [1.02–2.1], *p* = 0.04) were significant prognostic factors (*p* < 0.1).

The stage (stage 2: HR = 2.16; 95%CI = 1.23–3.78; stage 3: 4.35; 95%CI = 2.19–8.63) and HER2 metagene score classification (HR = 1.76; 95%CI= 1.09–2.86) remained significant in multivariate analysis (Table 2), and Figure 5 shows the 20-year breast cancer-specific survival estimates based on the metagene score classification. The median survival time was 90 months in HER2-high and 196 months in HER2-low patients (Log-rank test; *p* = 0.037) with a survival rate of 73.8% among HER2-low patients compared to 55.7% among HER2-high patients at 5 years.

### 2.5. Metagene Score in the Neoadjuvant Setting

The metagene scores were estimated in 1232 samples from patients subjected to NACT enrolled I-SPY (n = 988), NOAH (n = 156) and CHER-LOB (n = 88) trials. There were strong positive (OR = 1.70; 95%CI = 1.49–1.94) and negative (OR = 0.52; 95%CI = 0.44–0.60) correlations between the HER2 and Luminal metagene scores, respectively, with a response to therapy in all samples. The Basal metagene score was not associated with the response to therapy (OR = 0.93; 95%CI = 0.82–1.06). Among cHER2+ patients subjected to targeted therapy (n = 396), the HER2 metagene score remains a strong predictor (OR = 2.26; 95%CI = 1.74–2.98), and the Luminal metagene score remains a weaker but significant predictor (OR= 0.74; 95%CI = 0.56–0.99). In HER-negative samples, the Luminal metagene gene score is a strong negative predictor for response to NACT (OR = 0.42; 95%CI = 0.34–0.51). Table 3 resumes the logistic regression analyses, and Figure 6 shows the waterfall plots showing the distribution of response to therapy and the metagene scores in all breast cancer samples and in HER2-positive ones. Appendix A resumes the response rate according to each score quantile.

To estimate the clinical application of the HER2MtGx scores, logistic regression models were compared and the most informative model to predict pCR in 396 cHER2 patients subjected to HER2-targeted therapy was fitted using Luminal and Her2 scores (ANOVA). The final model was used to predict individual chances to achieve pCR and the ROC analysis showed a 74% AUC with an overall sensitivity of 77.5% and specificity of 66.7% at the best threshold (estimated probability of response) at 44.6% (Appendix A).

Based on the overall pCR rate of 43.2%, the estimated lower and upper CI of 99% was 37% to 49%, respectively. Thus, we considered patients as potential bad and good responders if the predicted individual chance of response was below or above the 99% CI, respectively. A total of 170 and 154 patients (82% of the total) could be classified as good and bad responders, respectively. The remaining patients were classified as regular responders. The predictive positive value (PPV) was 63.5% for good responders and only 21.4% of patients achieved pCR among bad responders, with a predictive negative value of 78.6%. The pCR ratio among regular responders was 41.7%. Comparing the performance of the HER2MtGx 31-gene signature with the well-established PAM50 classification (HER2-enriched as good responders and no-HER2-enriched classified as bad responders) including the 324 patients classified as good or bad responders and using the ground truth the presence of pCR, the HER2MtGx model has a significant gain in sensitivity and in negative predictive value (*p* = 0.01, Appendix A).

The probability of having a distinct outcome (pCR rate) was estimated in each one of the three responders classification (regular, low and high) compared to that observed in all patients with cHER2-positive tumors treated with neoadjuvant without target therapy (baseline). The baseline pCR rate was 25.5%. For the good responders, the posterior distribution indicates 100% confidence that good-responder patients achieve a more-than-two-times increment in pCR rate compared to the baseline. On the other hand, there is a 71,7% confidence that the bad-responder patients have a 20% reduction in the pCR rate compared to the baseline. For regular responders, there is a 93.8% confidence in a 40% increment in pCR rate compared to the baseline (Appendix A).

## 3. Discussion

HER2-positive tumors are associated with an adverse prognosis [9,10] and several targeted therapies have been developed to target HER2 signaling pathways with significantly increased disease-free survival and response to neoadjuvant and palliative treatment [11]. Nowadays, all patients carrying HER2-positive tumors are candidates for targeted therapy. Although the genetic and molecular mechanisms involved in HER2 aggressiveness are well-established, most patients with tumors classified as cHER2+ do not achieve survival benefits from the targeted therapy. The low probability of benefit may be due to intrinsic resistance or misclassification [3,12,13]. Our study demonstrated that the routine use of IHC and ISH techniques to select patients for HER2-targeted therapy is fairly accurate. Using a metagene score is an alternative to improve the selection of patients with higher benefit probability.

In this comprehensive analysis, a per-sample transcriptome transformation to robustly identify biologically meaningful clusters in breast cancer samples was used to identify gene signals that captured the fundamental genomic alteration associated with the HER2-enriched molecular profile—a core amplification in the long arm of chromosome 17 surrounding and affecting the ERBB2 gene. The HER2MtGx 31-gene assay was able to identify the *ERBB2* amplification and was a strong predictor of response to targeted therapy.

Multiple studies have investigated molecular signatures to predict response to HER2-targeted therapy [14,15,16,17,18]. The HER2DX prognostic score, which combines clinical/pathological features with four different gene signatures (immunoglobulin, cell proliferation, luminal differentiation, and HER amplicon modules) comprising 27 genes, is the most evaluated so far. It has been shown to be a prognostic and predictive tool for cHER2+ breast cancer [19]. The HER2DX gene signatures were based on the association of gene expression and prognosis.

HER2MtGx assay uses a distinct concept to identify the HER2 signature based on the molecular profile. Based on the assumption that Basal-like, Luminal, and HER2-enriched are distinct molecular subtypes, we applied a model that identified 31 genes with high signal associated with one profile but not with the others. There are only seven genes in common with HER2DX: four are shared with PAM50 [20] and OncotypeDX [21] signatures (*ERBB2*, *GRB7*, *ESR1*, and *BCL2*) and for the remaining three, *STARD3* is in the HER2 metagene, and *AGR3* and *AFF3* are in the Luminal metagene (Appendix A).

The use of predictive tools for response to therapy is the major challenge in cancer targeted therapy. For decision-making in real-world applications, it is crucial to estimate the probability of non-superiority for therapy de-escalation. We applied a Bayesian approach to estimate the probability of success regarding pCR rate among the HER2MtGx clusters. We observed that patients treated with target therapy and tumors in the cluster mgC3, representing tumors with higher HER2 score, have a significantly higher probability of achieving pCR than cHER2+ patients treated with standard NACT. On the other hand, patients with tumors in the clusters mgC1 and mgC2, which represent tumors with Luminal and Basal profiles, respectively, are prone to not benefit from the addition of targeted therapy to the neoadjuvant scheme.

Although this study has limitations due to its retrospective nature, the demonstration of the chance to respond to therapy based on a metagene score profile that generalizes from RNAseq to microarray technologies demonstrates the power of this platform. Furthermore, new targeted therapies have been applied to what has been called HER2-low profiles [22]. We have demonstrated that HER2MtGx can better select the real HER2-positive tumors, consequently reclassifying a couple of cHER2+ samples as HER2-low. This may have remarkable implications in the current HER2 therapy scenario [11].

The targeted therapy concept is based on delivering a drug that interferes with a specific somatic alteration. However, in many malignant neoplasms, particularly carcinomas, multiple somatic alterations can be identified at the cancer diagnosis due to the clonal evolution [23,24], and targeted therapy is acting in a subset of malignant cells at diagnosis. Developing tools that capture such heterogeneity may be useful for treatment optimization.

## 4. Materials and Methods

Data source. The breast cancer dataset from the TCGA project was used as the discovery dataset. All data are available at https://portal.gdc.cancer.gov/, accessed on 20 February 2024. The datasets were downloaded using the gdc-client API (manifests are available at (https://github.com/lab-tds/HER2MtGx_metagene.git, accessed on 12 March 2024)). The test datasets were used to analyze survival and response to neoadjuvant chemotherapy (NACT). The METABRIC dataset was used for survival analysis in cHER2+ samples and was obtained from the cbioportal repository [25]. The METABRIC cohort was firstly designed to investigate how gene copy number aberrations can reveal novel breast cancer subtypes, which they named integrative clusters (iCluster). They have demonstrated that iC5 is driven by a significant 17q12-q21 amplification affecting the *ERBB2* gene [26].

The series GSE194040 (I-SPY trial, accessed on 27 February 2024 [27]), GSE50948 (NOAH trial, accessed on 11 April 2024 [28]), and GSE66305 (CHER-LOB trial; [29]) were downloaded from the Gene Expression Omnibus (https://www.ncbi.nlm.nih.gov/geo/, accessed on 4 March 2024) using the GEOquerry library [30] in R, version 2.76.0, and they were used to make inferences on response to NACT. Those datasets include some clinical/pathological data matched to transcriptomic data (affymetrix microarray) in patients subjected to NACT with or without HER2-targeted therapy.

TCGA HER2 status reclassification. The 2023 ASCO-CAP guidelines for HER2 classification were adopted to reclassify TCGA samples as cHER+ [2]. Briefly, cHER2+ tumor was defined by (a) the positive (3+) protein expression in IHC or (b) a 2+ positive expression and a gene amplification by in situ hybridization (ISH) testing was considered if *ERBB2*/CEP17 ratio ≥2.0 and average *ERBB2* copy number ≥4.0 signals/cell or *ERBB2*/CEP17 ratio <2.0 and average *ERBB2* signals/cell ≥ 6.0.

Integrative clustering (iC10) classification in TCGA dataset. The iC10 subtypes were inferred in 1072 TCGA breast cancer samples [31]. The gene-based copy number data (log2 ratios) was used as the input to the iC10 package in R, version 2.0.2 [32]. The mean goodness of fit of the model was 95.5% (iC1 = 0.92, iC2 = 0.96, iC3 = 0.92, iC4 = 0.68, iC5 = 0.97, iC6 = 0.97, iC7 = 0.94, iC8 = 0.98, iC9 = 0.99 and iC10 = 0.90).

Unsupervised learning in the discovery dataset. A total of 1081 primary breast carcinoma samples from the TCGA dataset were used for gene expression data normalization. A per-sample standardization and asymmetric winsorization transformation were used for data normalization [8]. Briefly, the standardization step was applied to the RNAseq raw counts by centering and scaling them (z-counts). The smoothing step was then applied to leverage skewed transformations (smoothed-counts). The gene filtering step was applied after transformation using the default parameters with the awst library in R, version 1.16.0. A total of 10,214 genes were selected for clustering analysis.

All original cHER2+ and HER2 2+ by IHC were used for clustering analysis (287 potential HER2+ samples). Euclidean distance and Ward’s linkage method were applied to build the clusters. The Calinski and Harabasz curve (CH-curve) was used to infer the most reliable number of clusters [33], and the maximum Dunn’s index was used to decide among inferences from CH-curve [34].

Most informative gene selection. The transcript per megabase (TPM) normalized expression matrix from the potential HER2 samples was used. Samples were clustered into three distinct molecular subgroups (HER2, Basal, and Luminal). The main idea was to capture signals from genes that can reliably distinguish them. The analysis of variance (ANOVA) was applied, and the *p*-values were adjusted for multiple tests (false discovery rate—FDR). Genes with the adjusted *p* < 0.01 and the model R2 > 0.3 were selected.

To select the most informative genes for each molecular group, the regression coefficients were adjusted by a sigmoid function, limiting the top and bottom values to 3 and −3, respectively. The most informative genes selected were those with high coefficients in a group (coefficient > 2.7) and were non-positive in the others (coefficient ≤ 0). The selected genes composed the gene set for HER2, Basal, and Luminal metagene scores, and we are referring to this as the HER2MtGx 31-gene assay, or HER2MtGx from now on.

Statistical analyses. Chi-square test or Fisher’s exact test was used to analyze the association between categorical features. The *t*-test or the Wilcoxon rank sum test was used for association analyses based on their distribution for continuous variables. The ANOVA followed by the post hoc Dunn’s test, was used for association tests for more than two dependent variables. Survival analysis was right-censored at 20 years, and the Cox model was used to infer the coefficients associated with the survival for univariate and multivariate analyses. Logistic regression was used to analyze the metagene scores and the pathological response to therapy.

Logistic regression was used to estimate the coefficients of the HER2, Luminal and Basal scores associated with the response to NACT + targeted therapy. The study cohorts were used as strata to fit the models. The ANOVA test was used to compare the models’ performances. The Wald interval was used for approximation to estimate the confidence interval (CI) at 99% for the chance of response to NACT + targeted therapy considering the pCR as the success to infer the $\hat{p}$ and the $SE=\sqrt{\frac{\hat{p}\left(1-\hat{p}\right)}{n}}$. We considered patients as good or bad responders based on whether the individual chance of response was high (${\hat{p}}_{i}>\hat{p}+Z_{\alpha }SE$) or low (${\hat{p}}_{i}<\hat{p}-Z_{\alpha }SE$) compared with the established CI, respectively. The ROC curve using the probability to achieve pCR to NACT + targeted therapy as a predictor was used to estimate the AUC and to calculate the model performance parameters using the best threshold estimated by Youden’s J statistic.

Bayesian inference was performed using the bayesAB package for R, version 1.1.3. The prior knowledge about the response rate was based on patients with cHER2-positive samples in the NOAH trial (n = 51) treated with NACT without targeted therapy. The number of patients who achieved pCR was 13, and the likelihood function was a uniform Beta(α, β) function. The pCR rate was described as the Θ parameter (θA ~ beta(αA + sA, βA + fA) and θB ~ beta(αB + sB, βB +fB), where s = success (pCR) and f = failure (RD)) and random samples (10^5^) from each posterior distribution were drawn as a density function.

## 5. Conclusions

The HER2MtGx metagene is based on three linear subspaces representing the HER2, luminal and basal profiles and can reliably select patients for HER2-targeted therapy.

## Figures and Tables

**Figure 1 ijms-26-11809-f001:**
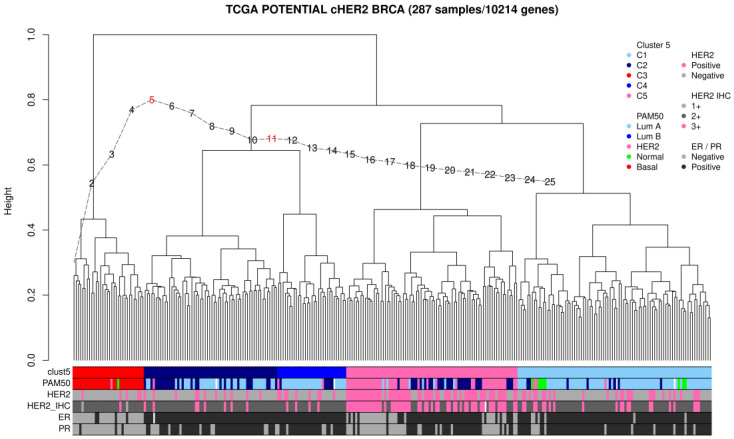
Unsupervised clustering including 287 potential cHER2+ samples from TCGA dataset. (Colored bar annotations: Cluster 5 = the inferred clusters by AWST pipeline; PAM50 = original TCGA molecular classification; HER2 = cHER2 reclassification; ER = estrogen receptor by IHC; PR = progesterone receptor by IHC). Dashed line represents the Calinski and Harabasz curve and the red numbers display the most reliable number of clusters. Missing color (white) represent non available data.

**Figure 2 ijms-26-11809-f002:**
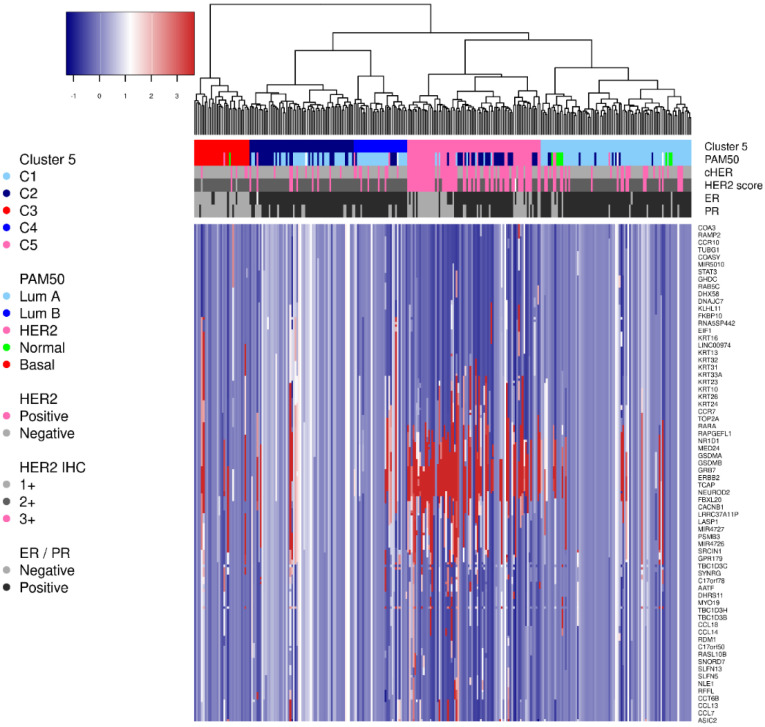
The unsupervised clustering of all potential cHER2-positive samples from TCGA was applied to the 17q12-q21 gene copy number (log2 ratio). Cluster 5 = the inferred clusters by AWST pipeline; PAM50 = original TCGA molecular classification; cHER2 = HER2 ASCO/CAP reclassification; HER2 score = HER2 IHC score; ER and PR = estrogen receptor and progesterone receptor by IHC.

**Figure 3 ijms-26-11809-f003:**
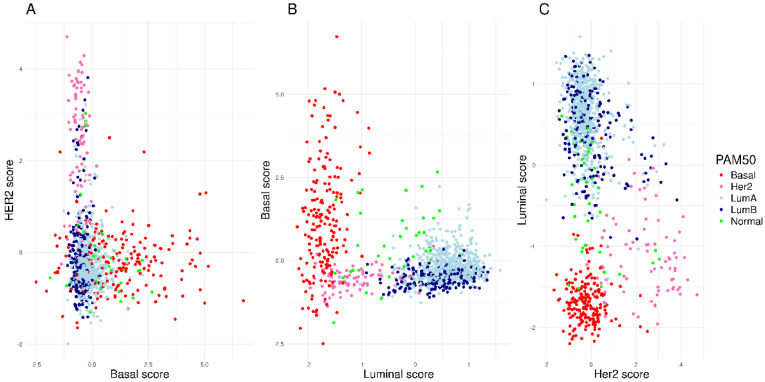
Distribution of the HER2MtGx scores in all TCGA breast cancer samples. The scatter plots (**A**–**C**) show the correlation among the HER2, Basal and Luminal scores and the PAM50 molecular profile.

**Figure 4 ijms-26-11809-f004:**
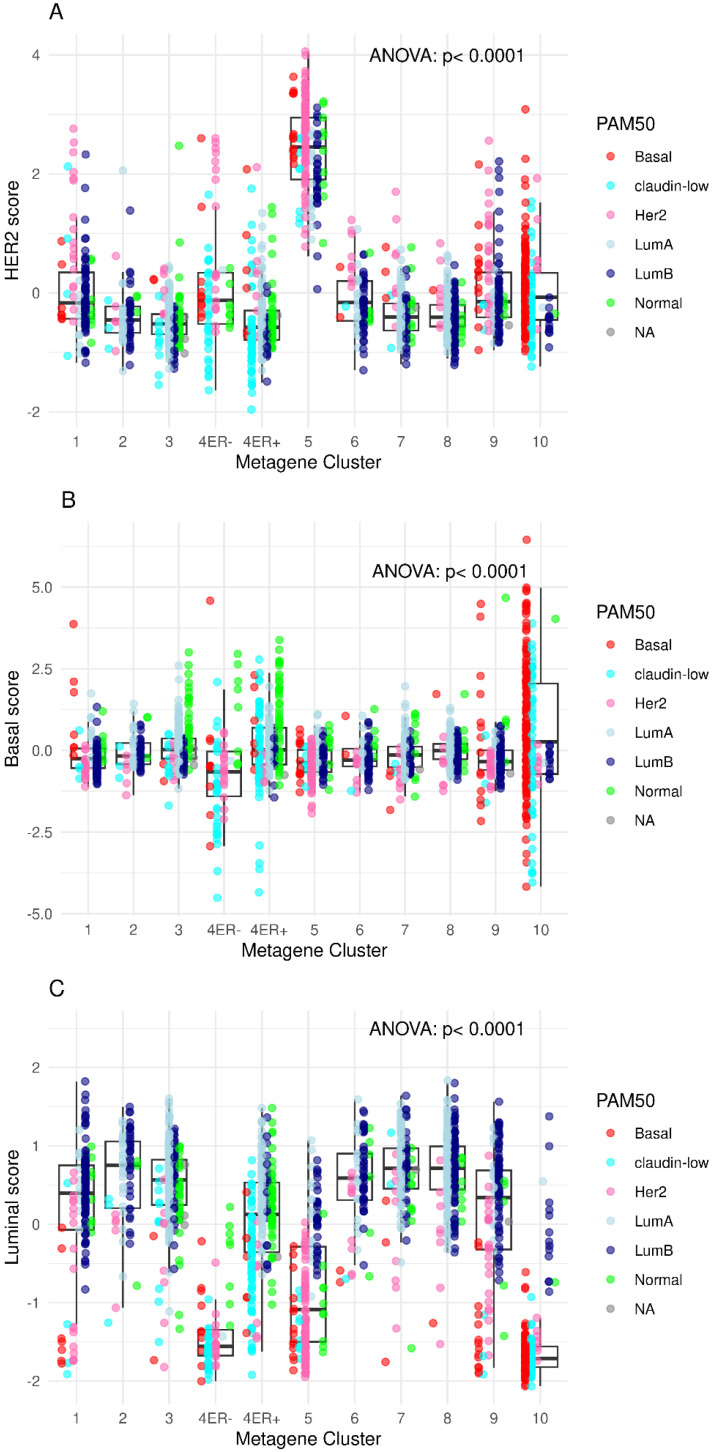
Boxplots showing the distribution of the (**A**) HER2, (**B**) Basal, and (**C**) Luminal metagene scores according to the integrative clustering and PAM50 classification in the METABRIC cohort.

**Figure 5 ijms-26-11809-f005:**
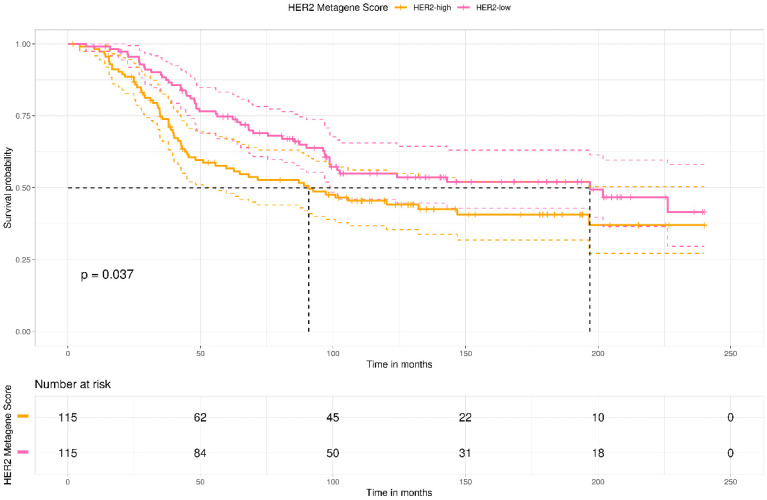
Kaplan–Meier estimates compare the HER2 high versus HER2 low scores (HER2MtGx) in all patients with HER2-positive non-metastatic breast cancer in the METABRIC cohort. Dashed lines represent the median survival time.

**Figure 6 ijms-26-11809-f006:**
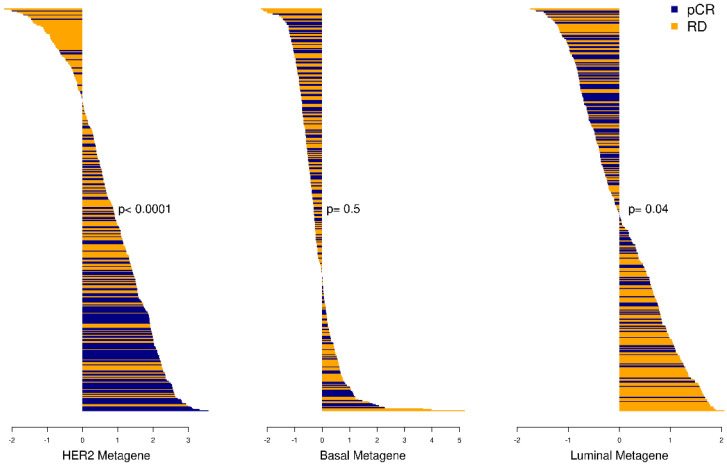
Waterfall plots representing the distribution of the response to neoadjuvant therapy in all patients with cHER2+ tumors in I-SPY, NOAH and CHER-LOB trial according to each HER2MtGx scores (HER2, Basal and Luminal).

**Table 1 ijms-26-11809-t001:** Clinical and pathological characteristics from all TCGA patients.

		HER2 Status (ASCO-CAP)		
Variable	N	Negative N = 984 ^1^	Positive N = 113 ^1^	*p*-Value ^2^
**Histology**	859			0.2
IDC		460 (59%)	56 (68%)	
ILC		126 (16%)	8 (9.8%)	
Mixed		81 (10%)	10 (12%)	
Other		110 (14%)	8 (9.8%)	
**Grade**	819			<0.001
1		173 (23%)	5 (6.2%)	
2		300 (41%)	36 (44%)	
3		265 (36%)	40 (49%)	
**ER status**	1046			0.042
Negative		204 (22%)	34 (30%)	
Positive		730 (78%)	78 (70%)	
**PR status**	1043			0.002
Negative		292 (31%)	52 (46%)	
Positive		638 (69%)	61 (54%)	
**HER2 IHC score**	622			<0.001
0		61 (12%)	0 (0%)	
1+		270 (53%)	2 (1.8%)	
2+		179 (35%)	20 (18%)	
3+		0 (0%)	90 (80%)	
**ERBB2 copy number (FISH)**	114	3 (2, 4)	60 (7, 125)	<0.001
**ERBB2 copy number/chr17 ratio (FISH)**	233	1.13 (1.05, 1.30)	2.60 (1.30, 4.05)	<0.001
**PAM50**	1083			<0.001
Basal		186 (19%)	5 (4.6%)	
Her2		36 (3.7%)	46 (42%)	
LumA		527 (54%)	34 (31%)	
LumB		187 (19%)	22 (20%)	
Normal		38 (3.9%)	2 (1.8%)	

^1^ n (%); Median (Q1, Q3). ^2^ Pearson’s Chi-squared test; Wilcoxon rank sum test; Fisher’s exact test for count data with simulated *p*-value (based on 2000 replicates).

**Table 2 ijms-26-11809-t002:** Multivariate Cox regression including all patients with HER2-positive tumors in METABRIC cohort.

Characteristic	HR ^1^	95% CI ^1^	*p*-Value
**Stage**			
1	—	—	
2	2.16	1.23, 3.78	0.007
3	4.35	2.19, 8.63	<0.001
**ER status**			
Positive	—	—	
Negative	0.90	0.55, 1.47	0.7
**HER2 metagene**			
HER2-low	—	—	
HER2-high	1.76	1.09, 2.86	0.022

^1^ HR = Hazard ratio, CI = confidence interval.

**Table 3 ijms-26-11809-t003:** Logistic regression analyses comparing the relationship between the HER2MtGx scores and pathological response to neoadjuvant therapy in I-SPY, NOAH and CHER-LOB trials.

	BRCA—All Samples			BRCA—cHER2+			BRCA—cHER2−		
Metagene Score	OR ^1^	95% CI ^1^	*p*-Value	OR ^1^	95% CI ^1^	*p*-Value	OR ^1^	95% CI ^1^	*p*-Value
**HER2**	1.70	1.49, 1.94	**<0.001**	2.26	1.74, 2.98	**<0.001**	1.20	0.87, 1.67	0.3
**Basal**	0.93	0.82, 1.06	0.3	1.10	0.83, 1.46	0.5	0.87	0.75, 1.02	0.084
**Luminal**	0.52	0.44, 0.60	**<0.001**	0.74	0.56, 0.99	**0.044**	0.42	0.34, 0.51	**<0.001**
**strata(study)**									
I-SPY	—	—		—	—		—	—	
CHER-LOB	0.88	0.52, 1.46	0.6	1.22	0.63, 2.35	0.5			
NOAH	1.01	0.68, 1.48	>0.9	2.14	1.09, 4.24	**0.027**	1.12	0.46, 2.54	0.8

^1^ OR = Odds ratio, CI = confidence interval.

## Data Availability

The data presented in this study are available on request from the corresponding author. Some data generated are confidential for patent purposes.

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
