# Peer review of "Int. J. Mol. Sci.2025, 26(24), 11809;https://doi.org/10.3390/ijms262411809"

_ijms, 2025, doi:10.3390/ijms262411809_

Round 1

Reviewer 1 Report

Comments and Suggestions for Authors

1.The study utilized multiple datasets with variations in sample sequencing platforms and treatment regimens, which may affect the robustness of the scoring system.
2.The manuscript mentions that HER2MtGx could identify HER2-low patients, but results and validation specific to this subgroup are lacking.
3.It is recommended to perform a head-to-head comparison of HER2MtGx with existing tools (e.g., HER2DX) on the same datasets regarding their performance in predicting pCR and survival outcomes.
4.The uneven distribution of sample sizes across different subgroups (e.g., 466 cases in mgC2 vs. only 25 cases in mgC3 for cHER2- patients) might introduce potential bias in the statistical results.

Author Response

We sincerely thank the reviewers for their thoughtful and thorough revision of our manuscript. Their insightful comments have substantially improved the quality and clarity of the work. We are glad to report that we have carefully addressed all concerns raised by both the reviewers and the International Journal of Molecular Sciences editorial office.

Comment 1: The study utilized multiple datasets with variations in sample sequencing platforms and treatment regimens, which may affect the robustness of the scoring system.

Answer 1: We thank the reviewer for this pertinent comment. The use of distinct platforms was intentional. The main idea here was to assess the model’s generalization capacity, which is crucial for ensuring its applicability across diverse datasets and experimental conditions. 

Comment 2: The manuscript mentions that HER2MtGx could identify HER2-low patients, but results and validation specific to this subgroup are lacking.

Answer 2: We thank the reviewer for this valuable observation regarding the model's application to identify HER2-low samples. Indeed, our statement referred to the HER2MtGx model's ability to mitigate potential misclassification arising from purely clinical or pathological inference. The HER2MtGx model is able to confidently identify the HER2 signature among cHER2 positive samples. Based on the observation in terms of survival and response to therapy, we 

Comment  3: It is recommended to perform a head-to-head comparison of HER2MtGx with existing tools (e.g., HER2DX) on the same datasets regarding their performance in predicting pCR and survival outcomes.

Answer 3: We appreciate this insightful suggestion and fully agree that comparative evaluation between models is critical.  Another reviewer also raised a similar point. However, since  HER2Dx is a commercial platform and the model coefficients are not publicly available, a direct comparison is currently not feasible. 

Notwithstanding, and as suggested by the other reviewer, we used the PAM50 classification as a control model. The PAM50 classification task does a similar job as the HER2MtGx as they were fitted to identify the molecular landscape associated with the amplification in EBBR2 gene. The major difference is the PAM50 is a categorical classification whereas the HER2MtGx provides a linear representation of the HER2 / Luminal / Basal domains. Your suggestion was crucial for us to decide to test how we can use the linear classification to improve response prediction. We decided to replace the k-means clustering, a nondeterministic algorithm that assumes the clusters are spherical, to a linear classification tuning the thresholds based on the statistical assumption of a Bernoulli distribution. The new results are in the main manuscript and we demonstrate this approach is a useful prediction model to use in clinical routine.

Comment 4: The uneven distribution of sample sizes across different subgroups (e.g., 466 cases in mgC2 vs. only 25 cases in mgC3 for cHER2- patients) might introduce potential bias in the statistical results.

Answer 4: We thank the reviewer for this observation. As mentioned previously, the k-means clustering is a nondeterministic algorithm and works well for spherical clusters. Thus, we decided to replace this analysis with a probabilistic model. All the modifications are highlighted in red in the main manuscript.

Reviewer 2 Report

Comments and Suggestions for Authors

Tiezzi et al. proposed a transcriptomic-based HER2MtGx 31-gene assay as a promising tool to identify HER2+ breast cancer tumors more likely to benefit from targeted therapy. The study is well-conceived and contributes an important advancement in refining predictive signatures for HER2+ disease. The statistical approaches appear sound and appropriate. this paper is suitable for publication.

One suggestion would be for the authors to provide additional discussion or, if possible, complementary analyses comparing the performance of the HER2MtGx 31-gene signature with established canonical classifiers—such as the PAM50 HER2-enriched subtype or HER2 amplification status alone—in predicting therapeutic response or survival outcomes. Such a comparison would help contextualize the predictive value of their proposed signature relative to existing benchmarks.

Author Response

We sincerely thank the reviewers for their thoughtful and thorough revision of our manuscript. Their insightful comments have substantially improved the quality and clarity of the work. We are glad to report that we have carefully addressed all concerns raised by both the reviewers and the International Journal of Molecular Sciences editorial office.0

Comment 1: Tiezzi et al. proposed a transcriptomic-based HER2MtGx 31-gene assay as a promising tool to identify HER2+ breast cancer tumors more likely to benefit from targeted therapy. The study is well-conceived and contributes an important advancement in refining predictive signatures for HER2+ disease. The statistical approaches appear sound and appropriate. This paper is suitable for publication.

Answer 1: We thank the reviewer for the positive comments regarding our original research manuscript. 

Comment 2: One suggestion would be for the authors to provide additional discussion or, if possible, complementary analyses comparing the performance of the HER2MtGx 31-gene signature with established canonical classifiers—such as the PAM50 HER2-enriched subtype or HER2 amplification status alone—in predicting therapeutic response or survival outcomes. Such a comparison would help contextualize the predictive value of their proposed signature relative to existing benchmarks.

Answer 2: We do agree with this point of view indicating that we should demonstrate how the HER2MtGx can improve the response prediction. In fact, the PAM50 classification task does a similar job as the HER2MtGx as they were fitted to identify the molecular landscape associated with the amplification in EBBR2 gene. The major difference is the PAM50 is a categorical classification whereas the HER2MtGx provides a linear representation of the HER2 / Luminal / Basal domains. Your suggestion was crucial for us to decide to test how we can use the linear classification to improve response prediction. We decided to replace the k-means clustering, a nondeterministic algorithm that assumes the clusters are spherical, to a linear classification tuning the thresholds based on the statistical assumption of a Bernoulli distribution. The new results are in the main manuscript and we demonstrate this approach is a useful prediction model to use in clinical routine. All the modifications are highlighted in red in the main manuscript.